



# Multiwavelength, aerosol lidars at Maïdo supersite, Reunion
# Island, France: instruments description, data processing chain
# and quality assessment
Dominique Gantois[1], Guillaume Payen[2], Michaël Sicard[1], Valentin Duflot[1,3], Nicolas
Marquestaut[2], Thierry Portafaix[1], Sophie Godin-Beekmann[4], Patrick Hernandez[2], Eric
Golubic[2].
[1] LACy, Laboratoire de l'Atmosphère et des Cyclones (UMR 8105 CNRS, Université de La Réunion, Météo-
France), Saint-Denis de La Réunion, France
[2] Observatoire des Sciences de l'Univers de La Réunion (OSU-Réunion), UAR3365, Université de La Réunion,
CNRS, IRD, Météo-France, 97490 Saint-Denis de La Réunion, France
[3] now at: Department for Atmospheric and Climate Research, NILU – Norwegian Institute for Air Research,
Kjeller, Norway
[4] LATMOS, Laboratoire Atmosphères, Observations Spatiales, UPMC, Paris, France
*Correspondence to*: Dominique Gantois (Dominique.gantois@univ-reunion.fr)
**Abstract.** Understanding optical and radiative properties of aerosols and clouds is critical to reduce uncertainties
in climate models. For over 10 years, the Observatory of Atmospheric Physics of La Réunion (OPAR) has been
operating three active lidar instruments (named Li1200, LiO3S and LiO3T) providing time-series of vertical
profiles from 3 to 45 km of the aerosol extinction and backscatter coefficients at 355 and 532 nm, as well as the
linear depolarization ratio at 532 nm. This work provides a full technical description of the three systems, details
about the methods chosen for the signal preprocessing and processing, and an uncertainty analysis. About 1737
night-time averaged profiles were manually screened to provide cloud-free and artifact-free profiles. Data
processing consisted in Klett inversion to retrieve aerosol optical products from preprocessed files. The
measurement frequency was lower during the wet season and the holiday periods. There is a good correlation
between the Li1200 and LiO3S in terms of stratospheric AOD at 355 nm (0.001-0.107; R = 0.92 $\pm$ 0.01), and with
the LiO3T in terms of Angström exponent 355/532 (0.079-1.288; R = 0.90 $\pm$ 0.13). The lowest values of the
averaged uncertainty of the aerosol backscatter coefficient for the three time-series are 64.4 $\pm$ 31.6 % for the
LiO3S, 50.3 $\pm$ 29.0 % for the Li1200, and 69.1 $\pm$ 42.7 % for the LiO3T. These relative uncertainties are high for
the three instruments because of the very low values of extinction and backscatter coefficients for background
aerosols above Maïdo observatory. Uncertainty increases due to SNR decrease above 25 km for the LIO3S and
Li1200, and 20 km for the LiO3T. The LR is responsible for an uncertainty increase below 18 km (10 km) for the
LiO3S and Li1200 (LiO3T). The LiO3S is the most stable instrument at 355 nm due to less technical modifications
and less misalignments. The Li1200 is a valuable addition to fill in the gaps in the LiO3S time-series at 355 nm or
for specific case-studies about the middle and low troposphere. Data described in this work are available at
https://doi.org/10.26171/rwcm-q370 (Gantois et al., 2024).
## 1. Introduction
Uncertainties concerning aerosol and cloud optical and radiative properties strongly affect surface climate and
also the accuracy in climate models (Hansen et al., 1997; Alexander et al., 2013). Aerosols can be of multiple





origins, compositions, sizes, and shapes, but can also interact at different temporal and spatial scales and be influenced by various dynamical processes. This makes their observation at the global scale and the modelling of their properties challenging. Improving our knowledge in this area implies to use different measurement techniques (in situ, active and passive remote sensing methods) synergistically and to provide continuous timeseries of high-resolution measurements in the low and middle atmosphere.

The Observatory of Atmospheric Physics of La Réunion (OPAR), located on Réunion Island near Madagascar, is currently equipped with more than 50 instruments distributed over three different sites: two historical coastal sites in the north, and a high-altitude site (Maïdo observatory, 2160 m asl, Baray et al., 2013), which now houses more than two-thirds of these instruments. OPAR is part of many international networks, including GAW (Global Atmospheric Watch), NDACC (Network for the Detection of Atmospheric Composition Change), SHADOZ (Southern Hemisphere Additional OZonesondes), and AERONET (Aerosol Robotic Network). Additionally, it is a part of the European research infrastructures ACTRIS (Aerosol, Clouds, and Trace Gases Research Infrastructure) and ICOS (Integrated Carbon Observing System).

Maïdo observatory (21.079°S, 55.383°E) is one of the very few active observational sites in the Southern Hemisphere (SH). It is scarcely influenced by anthropic aerosols. Its importance lies in the fact that the aerosol load in the atmosphere above Reunion Island is under the influence of many different sources of emission and dynamical processes responsible for short and long-range air-mass transports (Baray et al., 2013) such as biomass burning (BB) plumes (Edwards et al., 2006; Khaykin et al., 2020), which are emitted seasonally in the SH. Moreover, it is not rare for volcanic aerosols to be detected in the stratosphere above Maïdo observatory. In fact, several volcanoes are located at the same latitude (Hunga-Tonga), or in the same Hemisphere (Calbuco) as Reunion Island (Bègue et al., 2017; Khaykin et al., 2017; Tidiga et al., 2022; Baron et al., 2023; Sicard et al., 2023). The high altitude of this facility is also of great importance as it is located above the boundary layer during the night, allowing the observation of the free troposphere in a quasi-pristine environment.

Since its creation in 2012, the Maïdo facility has been equipped with four research lidar (light detection and ranging) instruments emitting electromagnetic radiations at different wavelengths. Three of them have been providing high resolution time series of aerosol extinction and backscatter vertical profiles in the UV (355 nm) and visible (532 nm) domains. As of today, these measurements have only been used occasionally for case studies (Bègue et al., 2017; Khaykin et al., 2017; Tidiga et al., 2022; Baron et al., 2023; Sicard et al., 2023). Full exploitation of these timeseries will enable to provide timeseries of aerosol extinction and backscatter profiles over Reunion Island. This can only be achieved after homogenizing the processing method for the three instruments.

This works provides a summary of the specifications of the systems and a full description of the preprocessing and processing methods used to produce different levels of the datasets for the three Maïdo lidars.

## 2. Instrumental description

**Table 1** is a summary of the characteristics of the three Maïdo lidars used to retrieve aerosol optical properties. A full description of each system is available in the following subsections.



| | Li1200 | | LiO3S | LiO3T | |
|---|---|---|---|---|---|
| **References** | (Dionisi et al., 2015; Vérèmes et al., 2019) | | (Portafaix et al., 2015) | (Duflot et al., 2017) | |
| **Time-serie** | In 2013-2017 | Added in 2017 | 2013-Current | In 2013-2017 | Added in 2017 |
| **Laser** | 2 × Quanta Ray **Nd**: YAG **pro-290** | | 1 × Quanta Ray Nd: YAG Lab 150 | 1 × Quanta Ray Nd: YAG Pro-290 | |
| **Emitted wavelength (nm)** | 355 | | 355 | 532 | 1064 |
| **Frequency (Hz)** | 30 | | 30 | 30 | |
| **Energy (mJ/pulse)** | 375 | | 150 | 250 | |
| **Reception channels (nm)** | Elastic 355M, 355H  Raman 387 | Elastic 355VL, 355L  Raman 387L | Elastic 355H, 355M  Raman 387M | Elastic 532$_{//}$, 532$_{\perp}$ | Elastic 532H, 1064  Raman 607 |
| **Telescope diameter (mm)** | 1 × 1200 | + 1 × 200 | 4 × 500 | 1 × 200 | + 1 × 500 |
| **Full overlap (km)** | ~ 15 | ~ 15 | ~ 4-5 | ~ 4-5 | ~ 4-5 |
| **Detectors** | Hamamatsu Photomultiplier tube (PMT) | | Hamamatsu PMT | Hamamatsu PMT | Photodiode (1064nm) |
| **Detector mode** | Photocounting | | Photocounting  Analog (355M) | Photocounting  Analog (532H, 1064) | |
| **Filter bandwidth (nm)** | 1 | 1.3 (355VL)  1.3 (355L)  3 (387L) | 1 | 1 | 0.7 (532H)  1.6 (607.7)  *4 (1064)* |
| **Raw vertical resolution (m)** | 15 | | 120 (2012 → 2017)  15 (2017 → current) | 7.5 | |
| **Acquisition** | Licel transient recorders | | | | |
| **Raw files integration time (minute)** | 1 | | 3 (2012 → 2017)  1 (2017 → 2022) | 2 | |
| **Reception channels (nm)** | 355H, 355M, 387 | + 355L, 355VL  + 387VL | 355H, 355M, 387 | Elastic // 532  Elastic ⊥ 532 | + 532H | + 607.7  *+ 1064* |
| **Observation capabilities (Range, km)** | 15-45 | 3-25 | 10-45 | 4-25 | 10-45 | 4-15 |

**Table1: Systems technical features. The letters VL, L, M and H after the wavelength stand for Very Low, Low, Medium and High, respectively. Only aerosol channels are listed here.**

## 2.1.    Lidar 1200 (Li1200)

The Li1200 is a Rayleigh Raman lidar able to measure vertical profiles of temperature between 30 and 100 km asl
and water vapor ratio from the ground up to 18 km (Vérèmes et al., 2019). Vertical profiles of aerosol light
extinction and backscattering can also be retrieved from the raw signals, as this instrument provides Rayleigh-Mie
scattering at 355 nm and Raman $N_2$ scattering at 387 nm. This instrument has been operating at the Maïdo facility
since 2012 and produces data since 2013.
*(i)    Actual configuration*
**The emission** consists in two Nd: YAG Quanta Ray pro 290 lasers, from Spectra-Physics, emitting
electromagnetic pulses at 1064 nm and 30 Hz. The final wavelength emitted is 355 nm, which corresponds to the
third harmonic of the initial wavelength. Each pulse delivers 375 mJ in 9 ns. The optical design of this lidar is
represented in **Figure 1**. The two laser beams are recombined through a polarizer cube, then sent to the telescope
through a series of mirrors. It should be noted that the lasers and the telescope are not in the same room, hence the
use of many mirrors. BE1 and BE2 lenses form an afocal of magnification 1.25, reducing the divergence of the



beams and mixing the phases. The goal is to reduce the hot spots, especially on the very fragile optic BE3. Last,
the laser beam is channeled through the center of the main telescope and magnified by a factor of 10 thanks to the
afocal system BE3 and BE4. The emission and main reception are therefore static coaxial, reducing the parallax
effect and the minimum overlap altitude.
**The reception** is made of two telescopes. The main telescope consists in a primary mirror of 1200 mm diameter
(M1200), which gave its name to this instrument. A secondary mirror HM sends the beam to the detection system.
The L1 lens allows the beam to converge faster, which explains the 3.6 m value of the focal length. GS1 is a glass
plate that sends about 8 % of the beam on the 355 nm Very Low (355VL channel) detector. As this detector is
located before the FD2 diaphragm, its field of view is the same as the one of the telescope, and it receives signal
in the very near-range. A density (ND) was placed in front of this detector to avoid saturation. FD2 is a diaphragm,
located at the focal plane of the telescope. Its aperture improves the geometrical factor of the telescope for the
detectors following it. DM1 is a dichroic filter that reflects 355nm and allows 387nm and 407nm to pass through.
GS2 is a glass plate that sends about 8% of the beam on the 355 nm Medium (355M) channel and 92% of the beam
on the 355nm Hight (355H) channel. DM3 is a dichroic filter which selects the 387 nm for the Raman N2 channel.
As of 2017, a second telescope, with a 200 mm M200 primary mirror and a focal length of 1 m, sends the signal
to a second detection box, using an optical fiber. This detection box filters the Rayleigh and Raman signals and
channels them respectively to the 355L and 387L detectors.
All the **detectors** are photomultiplier tubes (PMT) from Hamamatsu, reconditioned by the Licel company
(http://licel.com). The 355H, 355M, and 355L detectors are electronically shuttered to prevent saturation. The
**acquisition** cards also come from Licel and operate in photocounting mode. There are no analog channels. Raw
files follow a 1-minute integration.
**To summarize**, 355M and 355H channels exist since 2013, but their acquisition starts at 15 and 25 km,
respectively, to avoid saturation. Hence, the 355VL and 355L channels were added in 2017 to cover the first
altitude ranges below 15 km. The minimum height for 355L electronic shuttering is 450 m asl.
*(ii)    Previous modifications*
The detection unit was modified in 2017. Before that, the detection unit containing the 355L and 387VL
detectors did not exist. The M1200 mirror separation unit was modified. First, the part containing the FD1 to L3
optics, as well as the 355VL detector, did not exist. And there was an optic between IF2 and DM2 that would send
the visible signal to another detection unit. Indeed, originally, this lidar was supposed to operate at two emission
wavelengths, 355 and 532nm. However, during installation, due to mechanical and optical problems, only the 355
nm channel was retained (Dionisi et al., 2015).



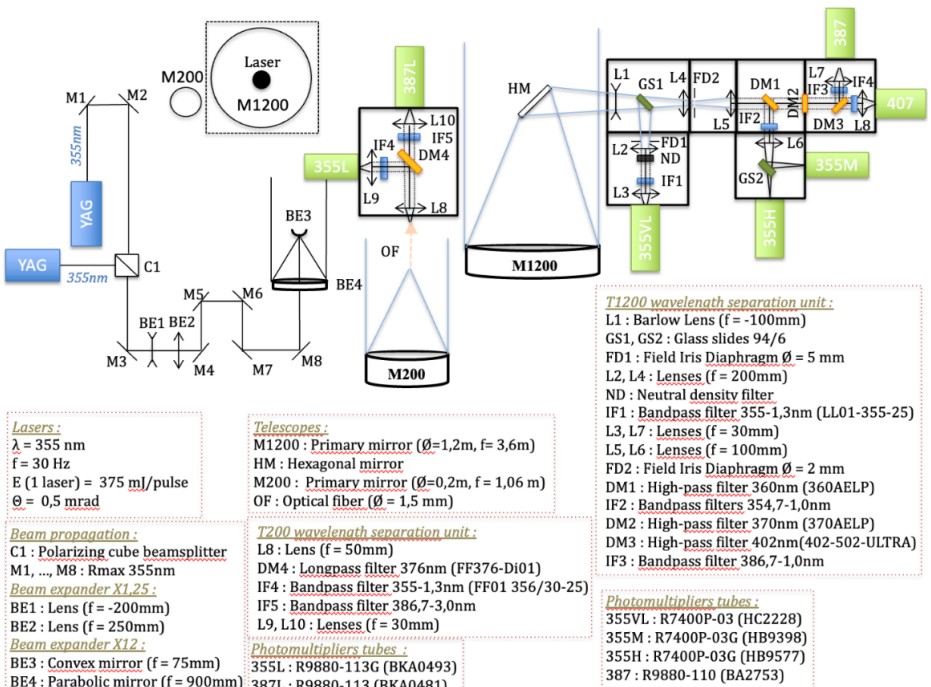


**Figure 1 : Li1200 optical scheme**
## 2.2. Stratospheric Ozone Lidar (LiO3S)
The Stratospheric Ozone Lidar (LiO3S) works with the DIfferential Absorption Lidar (DIAL) technique and
provides vertical profiles of ozone ($O_3$) concentration in the stratosphere, between the tropopause and about 45 km
(Godin-Beekmann et al., 2003; Portafaix et al., 2003). To this end, two different wavelengths are emitted: a 308
nm signal strongly absorbed by ozone molecules and a 355 nm signal weakly absorbed by ozone molecules.
Vertical profiles of aerosol light extinction and backscattering can be retrieved from the elastic scattering at 355
nm and Raman $N_2$ scattering at 387 nm. From 2000 to 2012, the LiO3S was located at the Moufia University
campsite in Saint-Denis and provided ozone vertical profiles. It was moved to the Maïdo facility in 2012 and has
been measuring from this location since 2013.
*(i) Actual configuration*
**The emission** set-up consists in two different lasers. A XeCl PulseMaster PM-800 Series excimer laser, from
LightMachinery, emits electromagnetic pulses at 308 nm wavelength with a frequency of 40 Hz and pulse energy
of 220 mJ. A Nd: YAG Lab 150 laser from Spectra-Physics emits electromagnetic pulse at a 1064 nm wavelength
with a frequency of 30 Hz. The final wavelength emitted by the Nd: YAG laser is 355 nm, corresponding to the
third harmonic of the emitted wavelength. The pulse energy at this wavelength is 130 mJ. The laser beam diameter
is about 10 mm, and its divergence is 0.5 mrad. The optical design of this lidar is represented in **Figure 2**. Again,
the emission and reception of this lidar are located in different rooms, explaining the use of many mirrors. The
expander consists in three lenses, BE1, BE2 and BE3, magnifying the signal by a factor 10. The final beam has a
100 mm diameter.



**The reception** is made of four 500 mm diameter telescopes. The primary mirrors are M1, M2, M3 and M4.
The signal is emitted at the center of these telescopes, and the distance between the emission and the center of each
telescope is 600 mm. At the receiving end, the signal is a focused from each telescope to a corresponding optical
fiber, which are positioned in line before entering the detection box. In this box, a diffraction grating separates the
different wavelengths. Internal mirrors allow the beam to be reflected in the detectors. Finally, a glass plate
discriminates the high and low energy channels at 355 nm.
All the **detectors** are PMT from Hamamatsu and the signal **acquisition** cards are from Licel. The 355 nm
detectors are electronically shuttered to avoid saturation.  The acquisition is in photocounting mode only for the
high energy channels, and in photocounting and analog mode for the low energy channels. Raw files follow a
1minute integration.
*(ii)   Previous modifications*
Before 2017, the electronic obturation concerned only 355H and 308H channels, and a mechanical chopper
shuttered 355M, 308M and Raman channels at the entrance of the detection box. In 2017, this chopper
malfunctioned and was replaced by electronic obturation for the 355M and 308M channel. Raman channels were
not shuttered anymore. The initial integration time was 3 minutes and was reduced to 2 and then 1 minute. During
this period, the vertical resolution was modified from 120 m to 15 m.

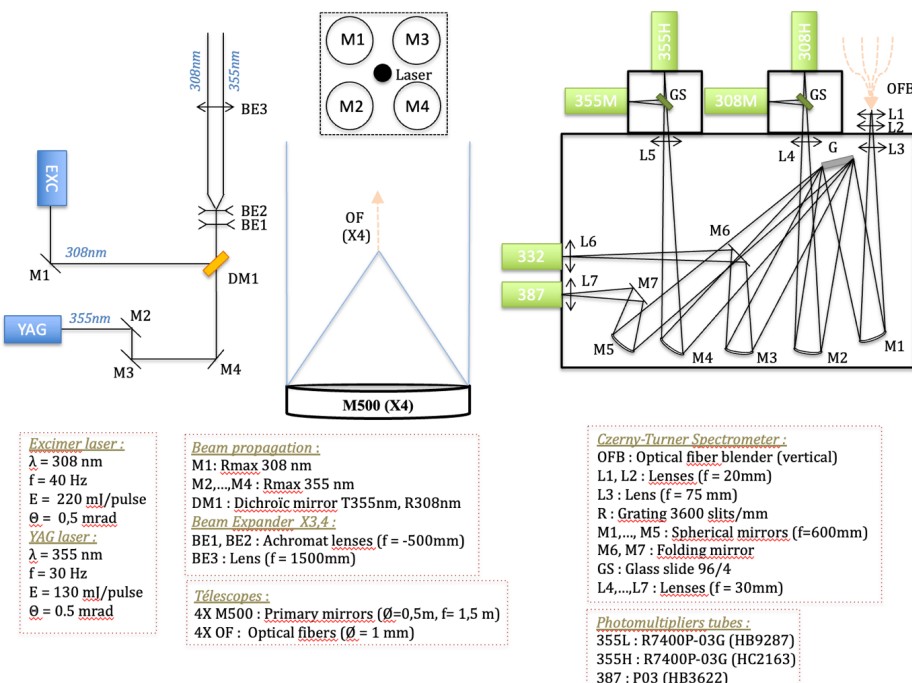


**Figure 2 : LiO3S optical scheme**


**2.3.   Tropospheric Ozone Lidar (LiO3T)**



The Tropospheric Ozone Lidar (LiO3T) also works with the DIAL technique and provides vertical profiles of
ozone ($O_3$) concentration in the troposphere, between 6 and 25 km (Duflot et al., 2017). To this end, two different
wavelengths are emitted using stimulated Raman scattering: a 289 nm signal strongly absorbed by ozone molecules
and a 316 nm signal weakly absorbed by ozone molecules. Vertical profiles of aerosol light extinction and
backscattering can be retrieved from the residual emission of the laser in terms of elastic scattering at 532 nm and
1064 nm, and Raman $N_2$ scattering at 607 nm. From 1993 to 2012, the LiO3T was located at the Moufia University
campsite in Saint-Denis and provided ozone vertical profiles. It was moved to the Maïdo facility in 2012 and has
been measuring from this location since 2013. The first aerosol dedicated polarized channels were installed in

172 2014.

*(i)      Actual configuration*
**The emission** consists in a Quanta Ray Pro 290 laser from Spectra-Physics emitting initially at 1064 nm at 30
Hz. While the fourth harmonic (266 nm) is used to retrieve tropospheric ozone profiles (through its passage in a
Raman cell generating 289 and 316 nm pulses), we use the second harmonic (532 nm) to retrieve aerosol light
extinction and backscattering. Each pulse at 532 nm provides an energy of 250 mJ. The laser beam diameter is of
about 10 mm, and its divergence is about 0.5 mrad. The optical design of this lidar for aerosol measurements is
represented in **Figure 3**. Again, the emission and reception of this lidar are located in different rooms, explaining
the use of many mirrors. The lenses, BE1, BE2 and BE3, magnify the signal by a 15 factor. The final emitted beam
diameter is 100 mm.
**The reception** is made of two telescopes: one for the Rayleigh and Raman channels (532, 607 and 1064 nm,
respectively), and the other for the polarized channels at 532 nm. The first telescope (M500) consists in a 500 mm
diameter primary mirror. An optical fiber located at its focal point, conducts the signal to the detection box.
Dichroic filters separate the 532, 607 and 1064 nm wavelengths. The second telescope consists in a 200 mm
diameter primary mirror immediately followed by a polarizing cube. An optical fiber leads the polarized and cross-
polarized beams to interference filters and to the detectors.
All the **detectors** are PMT from Hamamatsu, except for the 1064 nm detector, which is an avalanche diode
with a 3 mm diameter sensor. The 532 high energy channel (532H) detector is the only one electronically shuttered.
All the **acquisition** cards are from Licel. The acquisition of the 532 nm polarized channel as well as the 607 nm
channel are in photocounting mode. The acquisition of the 532H channel is in photocounting and analog modes,
and the acquisition of the 1064nm channel is only in analog mode. Raw files follow a 2-minute integration.
*(ii)     Previous modifications*
In 2014, the 200 mm telescope (M200) and the T200 wavelength separation unit were installed, allowing for the
first aerosol measurements with polarized channels. In 2017, one of the four 500 mm telescopes initially dedicated
to ozone measurements was used for aerosol measurements. A second detection box was added, enabling the 607
nm and 1064 nm channels acquisition.



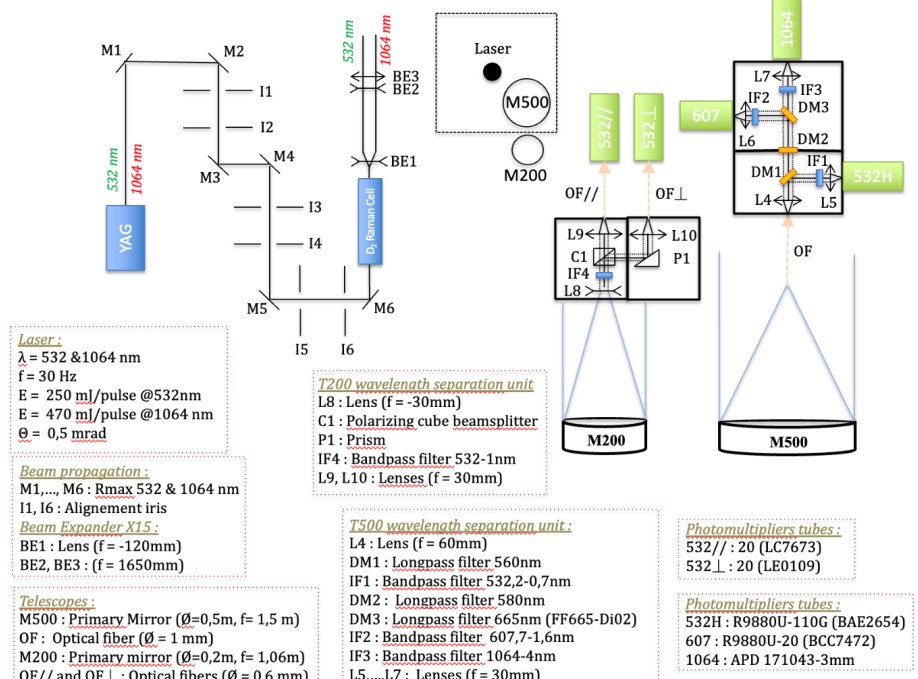

**Figure 3 : LiO3T optical scheme**

## 3. Routine measurements

The Maïdo lidars are research instruments that require manual handling and a constant human presence while operating. Maïdo observatory is a high-altitude facility (2160 m asl) and is located above the boundary layer in the free troposphere during the night. Acquisitions are only made during the night to increase the SNR. These instruments were originally intended to observe data in the stratosphere and the upper troposphere, so they are optimized to work at night, to improve the SNR up to very high in the atmosphere. That is why acquisitions are only made during the night. Measurements also require the absence of low-clouds or rain. The position of the Maïdo observatory on the west side of Reunion Island often protects the site from the clouds brought by trade winds. Notably, a ceilometer was installed at the Maïdo facility in 2019 and continuous observations revealed an average cloud frequency of respectively 20% and 40% during winter and summer nights (not shown).

Routinely, Maïdo lidars are operated two nights per week and measurements last from 7pm to 1am (local time, i.e. from 15 to 21 UTC). Specific campaigns (once or twice a year) can occasionally require to significantly increase the number of measurements. Operating these instruments implies to follow a strict, well-prepared protocol including basic check-ups and laser power control. A metadata file is routinely fed with technical specifics for each night of observation and after any instrumental modification. Automatization is currently in progress and could increase the frequency of routine measurements.

## 4. Data processing chain

### 4.1. Data processing levels



Our datasets follow a classification detailed in the following description. Data processing levels range from Level
0 to Level 2.
**(i)**  Level 0 products ($L_0$) are uncorrected and uncalibrated raw data files in Licel format at full
resolution produced by the instrument.
**(ii)**  Level 1 products ($L_1$) provide cloud-free data cleaned from any instrumental artifact (electronic
parasites, synchronization problems, power disrupt, etc.). The cloud mask is currently manual.
These corrections are essential for any user to be able to apply their own specific aerosol
preprocessing without errors linked to the instrument itself or the weather.
**(iii)**  Level 2 products ($L_2$) provide processed lidar data including: saturation correction, background-
sky correction, geometrical form factor correction and gluing between high and low-energy
channels. These products also provide the aerosol optical properties and their corresponding
uncertainties.
**4.2.**  **$L_0$ to $L_1$ processing chain**
Each instrument is equipped with an acquisition system provided by the Licel firm. The description of the
acquisition program producing output files in Licel format can be downloaded at
_http://licel.com/raw_data_format.html_. This process concerns three main sources of interferences: **(i)** Detection-
related interferences, **(ii)** Acquisition problems and **(iii)** Interferences linked to the lidar environment.
Any significative step of this process is tagged in the L1a output files to identify the corrections applied.
**4.2.1.**  **Detection interferences**
Detection-related interferences can generally be linked to electromagnetic disturbances, which can occur in
three different ways.
**(i)** An increased background signal concerning variable altitude ranges can impact the complete profile as
shown in **Figure 4a**. This disturbance affects one or several channels across a significant altitude range, making
the data acquisition unusable and requiring its withdrawal. This is one of the reasons files of a few minutes are
created. The strong disturbance in the signal enabled to fully automatize their detection. Notably, obturated
detectors are more sensitive to these disruptions. Experience proved that they are directly related to the use of cell
phones and Talky-Walkies. These instruments have been banned from the instrumental rooms during the
measurements, significantly decreasing the frequency of these cases.
**(ii)** A second electronic problem often encountered comes from electronic gating. In fact, if a high and low-
energy channel coexist, a peak can be observed on the low-energy channel raw signal, at the gated altitude of the
high-energy channel (**Figure 4b**). This parasite peak usually appears on 2 consecutive range bins. This type of
problem occurs when the detectors are obturated and can have a significant impact on the measurement. It is
therefore necessary to remove the corresponding values and replace them by an averaged value between the
previous and following range bins.
**(iii)** The third detection disturbance corresponds to a sudden peak of the signal on a single randomly located
range bin. They only concern LiO3S and LiO3T. The consequence on the nighttime averaged profile is shown on
**Figure 4c**. Generally, the intensity of these spurious peaks is consistent and significantly higher than the
atmospheric background noise. They are easily identified when the intensity of the received signal is much lower





and become negligible with a stronger signal. However, there is an intermediate zone where the intensity of the
received signal is close to the intensity of these peaks, making their detection more challenging. They are replaced
by an averaged value.

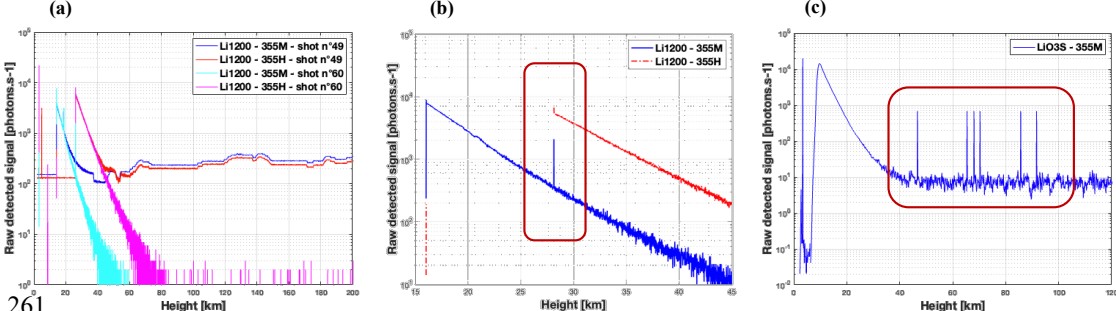

**Figure 4: (a) Raw Li1200 signal: background signal anomaly, (b) Raw Li1200 signal: peak from electronic gating, (c) Raw LiO3S nighttime averaged signal: random peaks in the far-range.**

### 4.2.2.    Acquisition problems

The acquisition program computes 1- or 2-minute integrated profiles, depending on the instrument. However,
with this acquisition program, the measurement cannot be stopped at the end of the current cycle. As a result, the
last file is generally shorter than the others and must be removed to guarantee consistent measurements.
Another issue was a time desynchronization of several minutes between the computer acquisition clocks in
2021, revealing a configuration default in the corresponding Network Time Protocol time servers. Time differences
could increase up to 15 minutes between the different computers. This default has been fixed and a time-correction
is applied for signals between 2012 and 2021.
Last, interaction between the different lidars working at the same time and emitting the same wavelength can
also lead to interferences and disturbances on sensitive channels. To avoid this issue, the lasers are synchronized
out of phase. However, errors with this offset can lead to files with a higher sky background than others. These
files are removed.

### 4.2.3.    Disturbance from clouds.

The SNR is most sensitive to the presence of low-altitude clouds. These clouds strongly absorb the emitted
photons and lead to high extinction levels and weak SNRs. They must be removed. High-altitude cirrus clouds can
also be removed if stratospheric aerosols are studied. Cloud-detection can be both automatic and/or manual. An
automatic detection of low clouds under 5 km height has been developed and can be used from 2019 up to now
using data from a Campbell CS135 ceilometer set up at the Maïdo facility in 2019. A manual cloud screening is
done for any remaining cirrus or low clouds. Automatization is in progress for this time-consuming work.

### 4.3.    $L_1$ to $L_2$ processing chain

The goal of this second processing chain is to retrieve vertical profiles of aerosol optical products. It involves
several key steps.

### 4.3.1.    Saturation correction



Saturation affects photomultiplier tube detectors with an acquisition card in photocounting mode. It concerns
the lower layers of the atmosphere and appears when the number of backscattered photons overcomes the capacity
of the acquisition card to discriminate them individually. Therefore, the backscattered signal is attenuated in the
corresponding layers. On the contrary, acquisition in analog mode is not affected by saturation, but has a weaker
SNR.
One solution is to combine (namely glue) analog and photocounting channels if both are available, which is
not always the case for our instruments.
The second option is to compare high and low-energy channels (or analog and photocounting channels if
available) in the lower layers and apply a dead-time correction to the photocounting channel using the Müller
equation. This is the solution we adopted for Maïdo lidars concerning aerosol, which is similar to what is done
for ozone and temperature processings (Leblanc et al., 2016a; Leblanc et al., 2016b). The dead-time parameter
($\tau_d$) corresponds to the minimum time for discriminating two consecutive photons. Our photocounting modes are
non-extensive, which means that the dead-time value is independent from the number of backscattered photons.
We then apply the Müller equation (Müller, 1973):

$$S_{desat} = \frac{S_{sat}}{1 - \tau_d \cdot \frac{c}{2 \cdot \delta_z \cdot L} \cdot S_{sat}} \quad (1)$$

With $S_{sat}$ (resp. $S_{desat}$) corresponding to the saturated (resp. desaturated) detected signal in number of photons
per second, $\delta_z$ the vertical resolution in meters, $c$ the light celerity in meters per second, and $L$ the number of shots.
A value of $\tau_d = 3.7 ns$ is chosen. This value is the one recommended by Licel manufacturers and was confirmed
after several experimental tests which are available in a summary document.

### 4.3.2. Background correction

The background sky signal ($S_{BC}$), is one of the main sources of noise affecting the SNR. It corresponds to: **(i)**
the detector noise, and **(ii)** the natural light emitted by the atmosphere and can be affected by the presence of the
moon during the night. The value of this signal is supposed to be constant with the altitude but in practice it
sometimes follows a linear variation due to the effect of the signal induced noise on the detector. Our instruments
are not equipped with any pre-trigger. Our method to calculate the ($S_{BC}$) value consists in performing a linear
regression or an averaging of the desaturated signal in an altitude range high enough to neglect the impact of the
backscattered signal compared to the ($S_{BC}$), typically between 80km and 120km.

### 4.3.3. Geometrical form factor correction

The overlap function $F(z)$ or crossover function is one of the major sources of uncertainties for ground-based
lidar measurements. It describes the fraction of the laser beam cross section contained by the telescope field of
view as a function of range. Its values vary between 0 (blind zone, no overlap) and 1 (full overlap). Originally,
Maïdo lidars were designed to study the high troposphere and the stratosphere and at these altitudes, the full overlap
is obtained, which is why there has not yet been a more specific study on these instruments.
Should this parameter not be corrected, the received lidar signal would be attenuated between the blind zone
and the full overlap, leading to incorrect optical values. Two approaches can be followed to determine this
parameter. **(i)** A theoretical calculation using equations found in Measures (1984) can be performed. However, it





implies the knowledge of several optical parameters which can vary over the timeseries, and different equations
must be used for coaxial and biaxial systems. **(ii)** The second and most common approach is experimental and
implies the use of horizontal measurements (Chazette et al., 2017). In fact, considering a constant and homogenous
atmosphere along the line of sight, a linear regression can be performed in an altitude range high enough to be far
from the full overlap. The difference between the logarithm of the signal and this linear regression gives an accurate
estimation of $F(z)$.
$$F(z) = \exp\left(\ln\left(S_2(z)\right) - y(z)\right) \quad (2)$$
With $S_2$ the desaturated, background corrected, and range corrected lidar signal, $y(z)$ the linear regression and
$z$ the altitude range.
It is physically impossible for these research instruments to measure horizontally. Therefore, the experimental
approach using vertical measurements (instead of horizontal) in aerosol-free conditions was performed to correct
overlap for the very low and low channels of the lidar 1200. As of today, no overlap correction was needed for the
LiO3S (full overlap under 10km) and LiO3T (full overlap between 3 and 4km).
**Figures 5a and 6a** reveal the variability of the overlap function over the time-series for both Li1200 VL and
L channels. This variability can be explained by slight misalignments of the lidar. Indeed, given the important
number of optical elements between the laser and the emission point, the risk of misalignment, even minor, is
significant. **Figures 5b and 6b** show the mean and standard deviation (std) of the overlap function from an
exponential regression. The small values of std are an indicator of a low-varying function, a result that allows to
use a unique overlap function rather than different functions for different periods. The estimated altitude of full
overlap was 10 km for the Very Low channel and 15k m for the Low channel.

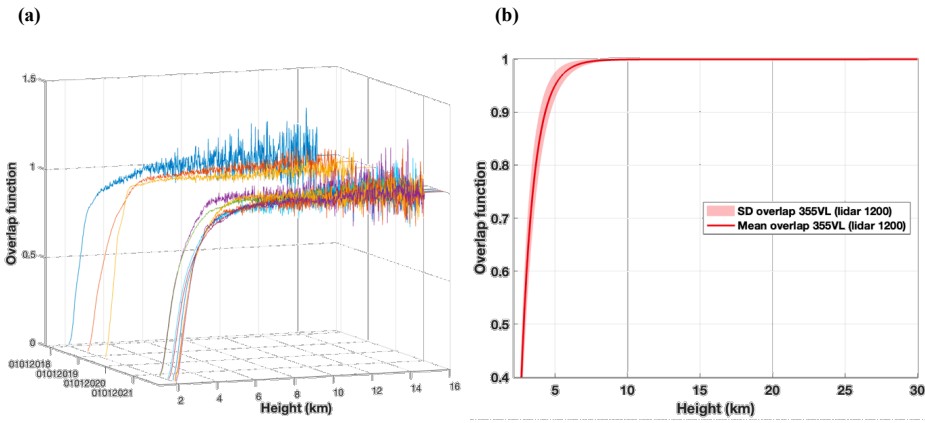


**Figure 5: Li1200 VL channel. (a) Time series of overlap functions, (b) Mean and standard deviation of**
**the overlap function.**



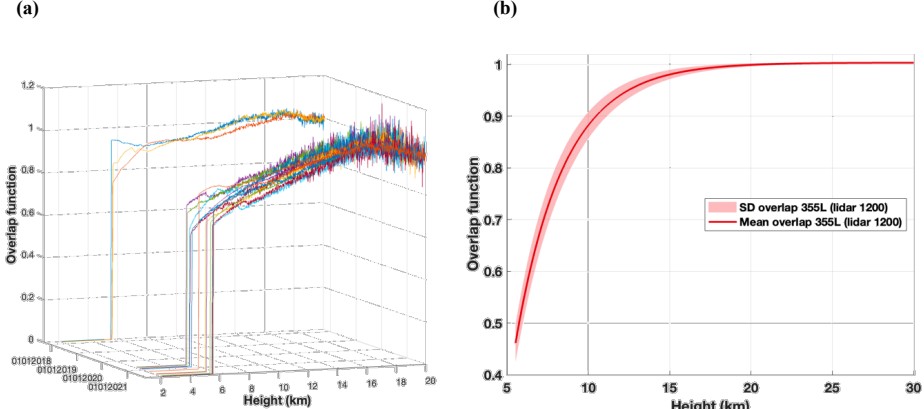

**Figure 6: Li1200 L channel. (a) Time series of measured overlap functions, (b) Mean and standard deviation of the exponential regression of the overlap function.**

### 4.3.4. Smoothing

Smoothing is applied on the lidar signal to increase the accuracy of the retrieved aerosol profiles. For the three time-series, smoothing was achieved using a low-pass filter with a Blackman window (Blackman and Tukey, 1958). The number of points for the filter was altitude-dependent and channel-dependent.

$$S_{filt}(z) = S_2(z)/F(z) * \frac{coef}{\sum coef} \quad (3)$$

$$coef(n) = 0.42 - 0.5 * cos\left(\frac{2\pi n}{W-1}\right) + 0.08 * cos\left(\frac{4\pi n}{W-1}\right), 0 \leq n \leq M-1 \quad (4)$$

With $S_{filt}$ the smoothed signal, $S_2$ the desaturated, background corrected, and range corrected lidar signal, $M$ half the length of the window and $W$ the weight of the filter.

**Figures 7a-c** represent the new vertical resolution for each channel of each instrument. Two methods can be used to estimate vertical resolution after smoothing: **(i)** Impulse response method and **(ii)** Digital Filter. The latter was chosen for these time-series. It involves the mathematical calculation of the filter transfer function, using a cut-off frequency at -3dB (NDACC_resolDF, (Leblanc et al., 2016)).

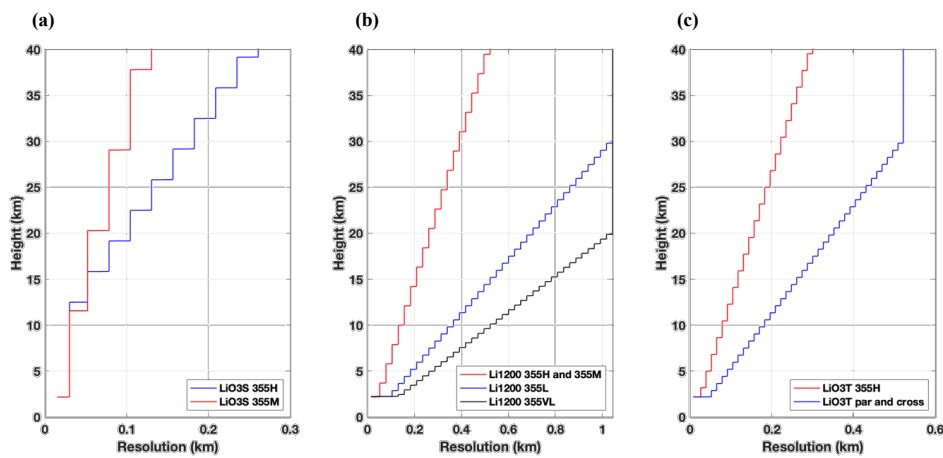

**Figure 7: NDACC vertical resolution of (a) LiO3S, (b) Li1200, and (c) LiO3T.**

### 4.3.5. Gluing near and far-range channels

High and low energy channels were combined for the LiO3S and the Li1200 using the gluing method of the square sinus and cosinus functions. The altitude range chosen for the gluing corresponded to a region where the high energy channel was not affected by electronic distortions and the low energy channel was not affected by too much noise.

$$
\begin{cases}
v1(z) = 0, & z < altmin \\
v1(z) = \sin^2\left(\dfrac{0 \xrightarrow{n} 1}{n} * \dfrac{\pi}{2}\right), & altmin \le z \le altmax
\end{cases} \quad (5)
$$

$$
\begin{cases}
v2(z) = 1, & z < altmin \\
v2(z) = \cos^2\left(\dfrac{0 \xrightarrow{n} 1}{n} * \dfrac{\pi}{2}\right), & altmin \le z \le altmax
\end{cases} \quad (6)
$$

With $n$ the number of range bins between $altmin$ and $altmax$, $v1$ the vector to apply to the high energy channel and $v2$ the vector to apply to the low energy channel.

The channels glued and used for inversion were: **(i)** 355VL + 355L + 355M + 355H and 355L + 355M + 355H and 355M + 355H for the Li1200, and **(ii)** 355H + 355M for the LiO3S. Each of these glued channels is available in the $L_{1b}$ files. Inversion was applied for each glued channels and corresponding optical products can be found in the $L_2$ files.

### 4.3.6. Calibration depolarization value for the LiO3T

Polarization channels enable to detect changes in the backscattered polarization state produced by the atmospheric particles. The laser provides quasi pure linear polarization. A polarizing cube beam splitter transmits the received linear polarized light and reflects the received cross polarized light. It is necessary to determine the polarization calibration factor before combining the two signals (Biele et al., 2000).

Three methods can be used: **(i)** Rayleigh calibration method (Behrendt and Nakamura, 2002), **(ii)** ±45° or Δ90° calibration methods (Freudenthaler, 2016), and **(iii)** 3 signals (total, cross and parallel) method (Reichardt et al., 2003). While methods 2 and 3 provide the smallest uncertainties, method 1 can be used retrospectively if no total channel existed. The apparent Volume Linear Depolarization Ratio (VLDR*) can then be calculated following:

$$
VLDR^* = \frac{K}{\eta^*} * \frac{S_r}{S_t} \quad (7)
$$

With $t$ and $r$ the respective transmitted and reflected parts of the signal $S$, $\eta^*$ the apparent calibration factor and $K$ the calibration factor correction parameter.

The VLDR can then be computed using the polarization crosstalk parameters for the transmitted and reflected signals ($G_{t,r}$ and $H_{t,r}$):

$$
VLDR = \frac{VLDR^*(G_t + H_t) - (G_r + H_r)}{(G_r - H_r) - VLDR^*(G_t - H_t)} \quad (8)
$$

The total signal will also be reconstructed following:

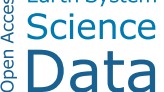


$$S_{total} = \frac{\frac{\eta^*}{K} H_r S_t - H_t S_r}{H_r G_t - H_t G_r} \quad (9)$$



The aerosol backscatter $\boldsymbol{\beta_a}$ will then be deduced from the total signal $\boldsymbol{S_{total}}$ using Klett inversion. The backscatter
ratio $\boldsymbol{R}$ will be calculated following:

$$R = \frac{(\boldsymbol{\beta_a} + \boldsymbol{\beta_{mol}})}{\boldsymbol{\beta_{mol}}} \quad (10)$$



Finally, the Particle Linear Depolarization Ratio ($\boldsymbol{PLDR}$) can be computed following:

$$PLDR = \frac{(1 + LDR_{mol}) * VLDR * R - (1 + VLDR) * LDR_{mol}}{(1 + LDR_{mol}) * R - (1 + VLDR)} \quad (11)$$



In our case, we used the Rayleigh method before 2017 and the 3 signals method after 2017. We used a linear
molecular depolarization ratio ($\boldsymbol{LDR_{mol}}$) of 0.00398 at 532nm (Behrendt and Nakamura, 2002) to estimate $\boldsymbol{\eta^*}$, and
a $\boldsymbol{K}$ factor of 1 to estimate $\textbf{VLDR}^*$. Crosstalk parameter values were considered ideal: $\boldsymbol{G_t = 1}$, $\boldsymbol{H_t = 1}$, $\boldsymbol{G_r = 1}$ and
$\boldsymbol{H_r = -1}$.
### 4.3.7.    Optical products: Klett inversion
This step is mandatory to retrieve aerosol optical properties from the detected lidar signals. However, it implies
to resolve an order 1 Bernoulli equation with several unknown parameters. Several methods exist such as: (i) One
or two-components Klett inversion (Klett, 1981, 1985), (ii) Raman inversion (Ansmann et al., 1990, 1992), and
(iii) a synergistic method using Klett inversion and sunphotometer measurements to evaluate the lidar ratio (Raut
and Chazette, 2007).
Because Raman channels have currently a very low SNR, they are not included in this work and the two-
component Klett inversion method was chosen for the three systems. It implies to determine an *a priori* constant
value of Lidar Ratio (LR) and a clean, aerosol-free zone in the atmosphere (Rayleigh zone). Details about the
elastic two-component algorithm from Klett are available in **Appendix A**.
The solution proposed in Appendix A is:

$$\beta(\lambda, z) = \beta_a(\lambda, z) + \beta_m(\lambda, z) = \frac{S_2(\lambda, z).\exp\{2.\int_{z'=z}^{z_{ref}} \left(\frac{LR_a(\lambda, z')}{LR_m(\lambda, z')} - 1\right).\alpha_m(\lambda, z')dz'\}}{\frac{S_2(\lambda, z_{ref})}{\beta(\lambda, z_{ref})} + 2.\int_{z'=z}^{z_{ref}} LR_a(\lambda, z').S_2(\lambda, z').\exp\{2.\int_{x'=z}^{z_{ref}} \left(\frac{LR_a(\lambda, x')}{LR_m(\lambda, x')} - 1\right).\alpha_m(\lambda, x')dx'\}.dz'} \quad (12)$$

With $\boldsymbol{a}$ (resp. $\boldsymbol{m}$) the particular (resp. molecular) contribution, $\boldsymbol{\alpha(\lambda, z)}$ (resp. $\boldsymbol{\beta(\lambda, z)}$) the summed molecular and
particular extinction (resp. backscatter), and $\boldsymbol{LR}$ the Lidar Ratio. $\boldsymbol{S_2}$ corresponds to the range-corrected, sky
background corrected and desaturated signal. However, the signal used in this study for the inversion algorithm is
smoothed as explained in paragraph 4.3.4. and could be glued (Li1200, LiO3S) or recombined (LiO3T).
Several unknown parameters must be determined:

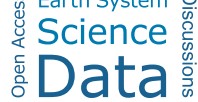

(i)    To retrieve the $LR_a$, we chose a constant LR value of 50 sr for the three instruments to be consistent

between the time-series and to target the most frequent aerosol types. Moreover it enables easier

comparisons with satellite data such as CALIOP products (Cattrall et al., 2005).

(ii)    The equation used to retrieve the molecular extinction was (Bates, 1984):
$$\alpha_m(\lambda, z) = \frac{P}{k * T} * \frac{4.02.10^{-28}}{\lambda^{4+(0.389\lambda + 0.09426\lambda^{-1} - 0.3228)}} \quad (13)$$

With $k$ corresponding to the Boltzmann constant. Atmospheric pressure $P$ and temperature $T$ were

retrieved from the Arletty AERIS product (*https://www.aeris-data.fr/*), relying on data from the

European weather forecast model ECMWF (European Centre for Medium-Range Weather Forecasts),

and producing interpolated data every 6h around Maïdo observatory (Hauchecorne, n.d.).


The molecular backscatter was then computed following:


$$\beta_m(\lambda, z) = \alpha_m(\lambda, z) * \frac{3 * K_f}{8\pi} \quad (14)$$

The King factor's value ($K_f$) is considered equal to 1 (King, 1923), and $\frac{3}{8\pi}$ corresponds to the $LR_m$.

(iii)    The last step was to determine for each daily measurement and each channel a reference 'Rayleigh'

zone $z_{ref}$ supposed free of any aerosols.


### 4.3.8.    Raman and 1064 nm channel issues

Klett inversion brings the problem of considering a lidar ratio constant with height. In fact, a single aerosol
plume is often made of several layers of particles with heterogenous backscattered lidar signals. Raman inversion
is one solution to deduce a vertical profile of lidar ratio from elastic and Raman channels. However, our Raman
channels have a poor SNR and are not usable for stratospheric or high tropospheric aerosols. The retrieval of
aerosol optical products using Raman inversion for low-energy channels (low and middle troposphere) is still
ongoing. There is also a misalignment issue for the 1064-nm channel leading to a poor SNR. This channel is
currently unexploitable.

## 5. Quality assessment

### 5.1.    Database statistics

A total of 1737 nighttime measurements were preprocessed between 2013 and 2023: 710 files for Li1200, 534
files for LiO3S, and 493 files for LiO3T. Notably, the mean percentage of rejected files was higher for Li1200
(52.7%), than LiO3T (44.8%) and LiO3S (32.7%). **Figure 8** shows the cumulated monthly number of validated
$L_2$ profiles for each instrument, the monthly mean number of rejected files and corresponding tags (cloud detection,
technical issue, low SNR). It should be noted that most observations were made during the May to November
period (austral winter, dry season) compared to the December to April period (austral summer, wet season), which
is consistent with the higher cloud and rain occurrence during the wet season. The mean percentage of validated
$L_1$ files was 62.4% during the dry season and 48.5% during the wet season. The lower frequency of measurements





in January, July, August, and December also concurs with two important holiday periods. The frequency of
technical issues and lower SNR is statistically higher during the months with a greater number of measurements.

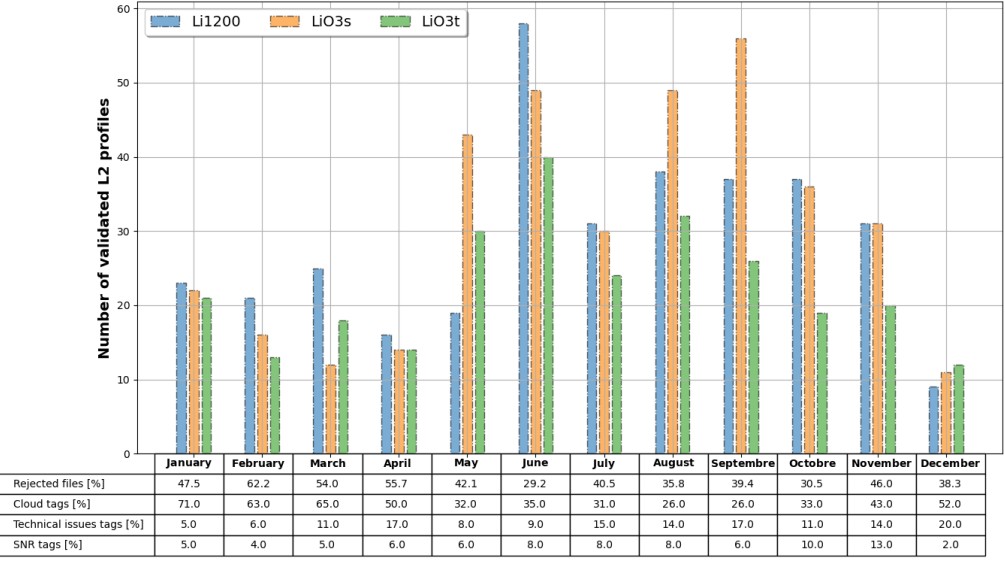

| | January | February | March | April | May | June | July | August | Septembre | Octobre | November | December |
|---|---|---|---|---|---|---|---|---|---|---|---|---|
| Rejected files [%] | 47.5 | 62.2 | 54.0 | 55.7 | 42.1 | 29.2 | 40.5 | 35.8 | 39.4 | 30.5 | 46.0 | 38.3 |
| Cloud tags [%] | 71.0 | 63.0 | 65.0 | 50.0 | 32.0 | 35.0 | 31.0 | 26.0 | 26.0 | 33.0 | 43.0 | 52.0 |
| Technical issues tags [%] | 5.0 | 6.0 | 11.0 | 17.0 | 8.0 | 9.0 | 15.0 | 14.0 | 17.0 | 11.0 | 14.0 | 20.0 |
| SNR tags [%] | 5.0 | 4.0 | 5.0 | 6.0 | 6.0 | 8.0 | 8.0 | 8.0 | 6.0 | 10.0 | 13.0 | 2.0 |


**Figure 8: Number of validated files for the three instruments in the period 2013-2023. In the table below, mean percentage of rejected files and tagged files for each month.**
### 5.2.    Instrumental capabilities
The gluing technique allowed to determine different altitude ranges for each lidar depending on the channels
available. **Table 1** provides a summary of the theoretical instrumental performances in terms of altitude ranges.
Apart from the number of glued channels, other parameters can influence the maximum altitude (SNR) or the
minimum altitude (Overlap, SNR) of the validated $L_2$ vertical profile. The LiO3T at 532 nm is ideal to investigate
the low and mid troposphere. The high troposphere and stratosphere can be studied at 355 nm (Li1200 and LiO3S)
or 532 nm (LiO3T – from 2017 until now).
### 5.3.    Instrumental intercomparison
In this study, we performed a comparison between the three instruments to detect any major discrepancies
using the Stratospheric Aerosol Optical Depth (sAOD) between 17 and 30 km. **Figure 9** displays the time-series
of sAOD at 355 nm (Li1200 and Li03S) for concomitant measurements and corresponding uncertainties. There is
a good overall consistency between the two instruments. The differences between the two time-series could be the
consequence of technical modifications (channel addition, optimization, misalignments). Three peaks periods of
high sAOD values can be identified: the emission of volcanic aerosols in the stratosphere during the Hunga-Tonga
eruption in 2022 (Kloss et al., 2022; Baron et al., 2023; Sicard et al., 2023), the Calbuco volcanic eruption in 2015
(Bègue et al., 2017) and the Australian bushfires in 2020 ( Khaykin et al., 2020). Higher differences in 2021 could
be the consequence of repeated misalignments for the Li1200.

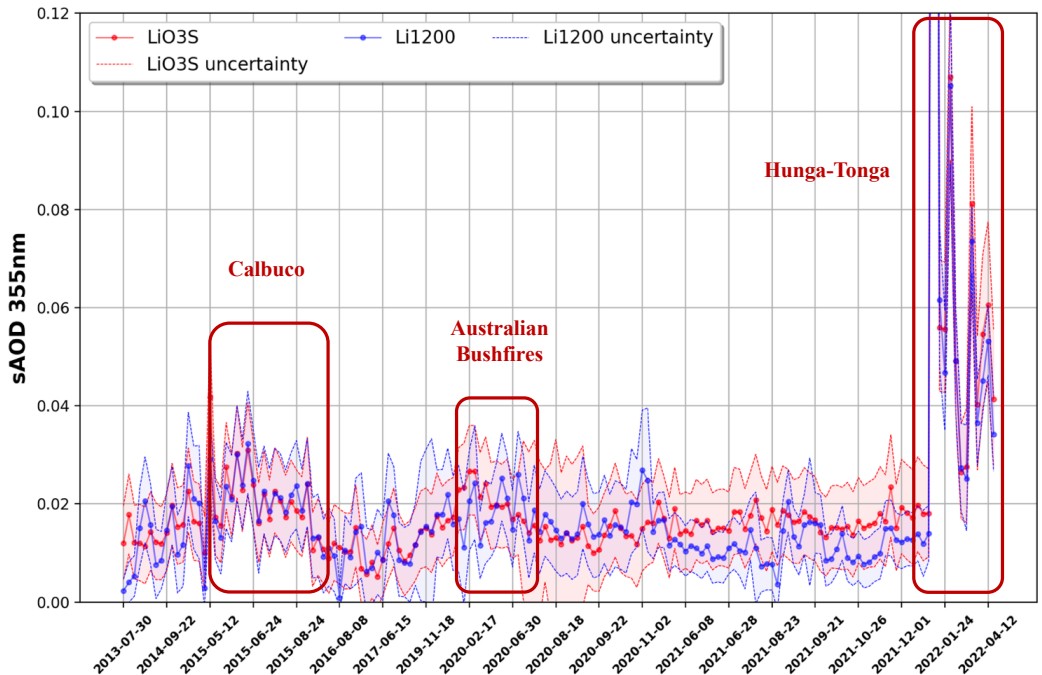

**Figure 9: Nighttime AOD (17 to 30 km layer) at 355 nm, from the Li1200 (red) and LiO3S (blue) (concurrent measurements) with corresponding uncertainties (dashed colored lines). Exceptional events circled in red. The horizontal timeline is not linear: one date out of eight is represented for visual purposes.**

The dispersion of sAOD values is represented in **Figure 10**. The sAOD at 355 nm varies between 0.001 and 0.107 for LiO3S and Li1200, with a mean of $0.019 \pm 0.012$ and $0.017 \pm 0.012$, respectively. A good correlation is found between the two lidars (correlation R = $0.924 \pm 0.005$).

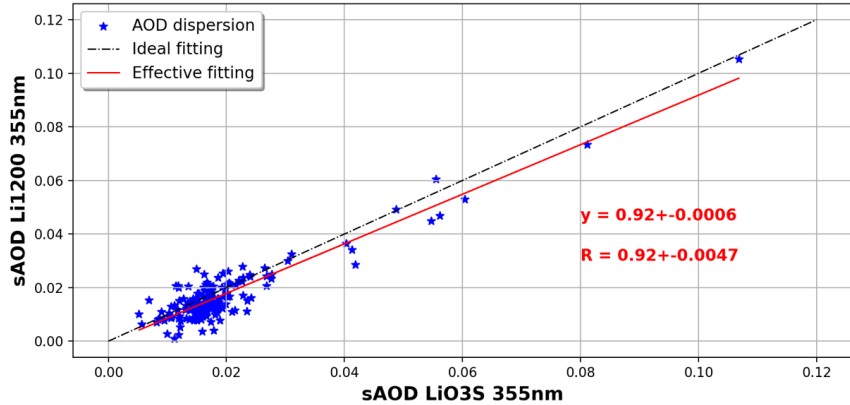

**Figure 10: Dispersion of the AOD (17 to 30 km layer) at 355 nm, between the Li1200 and LiO3S. The red line represents the theoretical linear regression.**

The correlation between the two instruments at 355 nm in terms of extinction values is higher above 17 km but lower from 10 to 17 km (**Appendix D**, **Figure D1**). In fact, for the Li1200: **(i)** low energy channels were added in




2017, **(ii)** there were changes in the minimal altitude of detection for the 355M channel, and **(iii)** this instrument
had many misalignments and underwent several optical upgrades, leading to modifications of the overlap function.
For further retrospective trend studies, it is important to note that the LiO3S has been the most stable instrument
throughout the time-series and is considered the reference instrument at 355 nm. However, data from the Li1200
can be used to fill the gaps of the LiO3S database depending on the altitude range targeted, but also for specific
case studies with the need to retrieve optical products for the middle and low troposphere.
The same analysis was performed for the LiO3T. To compare the two wavelengths, Ångström exponents (AE)
were computed between the LiO3T (532 nm) and alternatively the LiO3S (355 nm) and Li1200 (355 nm). **Figure**
**11** shows the dispersion of AE values. The order of magnitude of AE values varies between 0.0794 and 1.288 with
a mean of $0.56 \pm 0.29$ and $0.54 \pm 0.28$, respectively. Again, a good correlation is found between both datasets
(R = $0.901 \pm 0.128$). These values also demonstrate the variability of stratospheric aerosol size distribution
between 17 and 30 km (Gobbi et al., 2007; Burton et al., 2012).

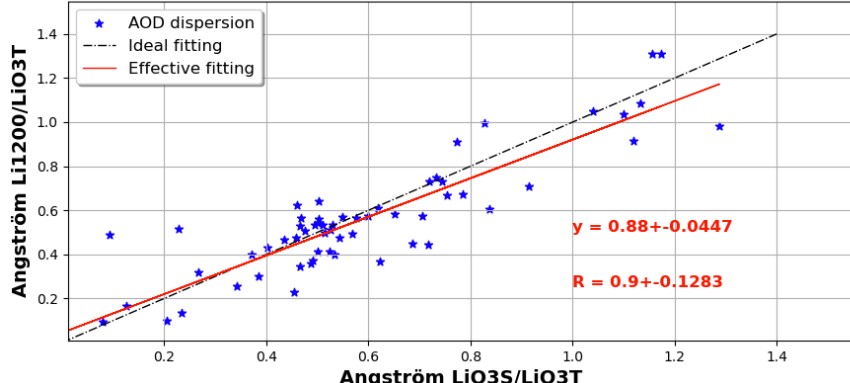


**Figure 11: Dispersion of the AE (17 to 30 km layer) between 355 and 532 nm. The black line represents the theoretical linear regression and the red line the actual linear regression.**

**5.4.      Main sources of uncertainties**
The total uncertainty budget of each lidar is described in **Appendix B**. Four sources of uncertainty were
propagated in quadrature (Sicard et al., 2009; Rocadenbosch et al., 2010): **(i)** uncertainty due to the Rayleigh
calibration value ($u_{altref}$), **(ii)** uncertainty due to the lidar ratio value ($u_{LR}$) with a distinction between *LR, top* and
*LR, bottom* defining the respective upper and lower error bars, **(iii)** uncertainty due to the SNR vertical distribution
($u_{SNR}$), **(iv)** and uncertainty due to the SNR value at the calibration altitude ($u_{SNR,altref}$). **Figures 12a-12c**
represent for three case reports the importance of each uncertainty relatively to the total backscatter in percentage,
and **Figures 12d-12f** represent the corresponding propagated total backscatter uncertainty for the three
instruments.
In **Figures 12a-12c,** the behavior of the uncertainties $u_{altref}$ (blue curves) and $u_{SNR,altref}$ (green curves) is
stable over the different altitude ranges. Notably, $u_{altref}$ comes from the 5% uncertainty of the molecular





backscatter, which determines the lower threshold for the total uncertainty. The $u_{SNR}$ uncertainty (purple curves)
is strongly influenced by the altitude, with minimal values at lower altitude ranges where the lidar signal is stronger,
and values increasing with the altitude. In fact, lidar signals are filtered before inversion, making $u_{SNR}$ the
predominant error at higher altitude levels. Oppositely, the $u_{LR}$ uncertainty (orange and yellow curves) is the lowest
at the calibration altitude and increases in the lower levels, where it becomes predominant. The systematic
uncertainty on the LR value was set to 30% for this study. Therefore, the total uncertainty is the lowest in mid-
altitude ranges before increasing in lower and higher altitude levels. Sharp spikes in $u_{LR}$ can be observed just
below 20km for the LiO3S and Li1200, and below 8 km for the LiO3T. They are linked to the presence of aerosol
plumes and emphasize the impact of aerosols on the uncertainty values in lower altitude levels.
For the LiO3S (H+M glued channel), the total relative uncertainty reaches 15% at 10 km, decreases down to
6% around 20 km, and increases up to 8% around 35 km. (**Figure 12a**). Without the aerosol layer, the minimum
error would be reached around 15 km. For the Li1200 (H+M+L+VL glued channel), the total relative uncertainty
reaches 20% at 7 km and decreases down to 5% from 20 km up. (**Figure 12b**). The uncertainty due to the SNR is
very low compared to the LiO3S, as this instrument is designed to reach very high-altitude levels, and the signal
used for inversion is made of four filtered signals with complementary vertical capacities. Without the aerosol
layer, the minimum error would be reached around 17 km. For the LiO3T, the total relative reaches 10% at 4 km,
decreases down to 6% around 8 km, and increases up to 20% around 25 km. (**Figure 12c**). The uncertainty due to
the SNR is higher than the previous lidars because this instrument is designed for tropospheric measurements.

Earth System
Science
Data


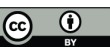



**Figure 12: Upper row: random cases showing the molecular backscatter (black), the backscatter coefficient (blue)**
**and its apparent uncertainty (red dotted line) for the (a) LiO3S (25/01/2022), (b) Li1200 (25/01/2022) and (c)**
**LiO3T (25/09/2017). Lower row: corresponding relative uncertainties for the (d) LiO3S, (e) Li1200 and (f) LiO3T.**

In **Figures 12d-12f**, the three instruments demonstrate their capacity to detect aerosol layers with relatively

low error rates and a high resolution. **Figures 12d-12e** specifically show their ability to identify variations within
the aerosol layer between 18 and 20 km. For the LiO3T (**Figures 12f**), the aerosol layer between 4 and 8 km is
exceptionally well defined, with relatively low error values. Apart from these aerosol layers, the molecular




backscatter (in black) tends to align closely with the uncertainty of the total backscatter (in red). In fact, background
aerosols are characterized by very low backscatter and extinction values, leading to the relatively high sAOD
uncertainties observed in **Figure 9**: higher for background aerosols but lower for cases with a stronger aerosol
load, such as from Australian fires or volcanic aerosols. Focusing on the uncertainty specific to aerosol backscatter
(rather than the total) is essential to improve the uncertainty analysis, along with a statistical analysis of the dataset
to minimize disruptions caused by transient aerosol events. Time-series of aerosol backscatter relative total
uncertainties were computed for the three instruments and the corresponding mean and standard deviation are
represented **Figures 13a-c**. Values are high and easily reach 100% for the three instruments because of the very
low values of aerosol backscatter coefficients above Maïdo observatory. The mean uncertainty is the lowest for
the LiO3S between 18 and 25 km (64.4± 31.6 %). It increases under 18 km and above 25 km with relative
uncertainty values reaching more than 100% due to the very weak aerosol backscatter values at these altitude
ranges. The mean uncertainty for the Li1200 is also the lowest between 18 and 25 km (50.3 ± 29.0 %). It increases
under 18 km and above 25 km with relative uncertainty values relatively lower than the LiO3S due to a lower
SNR, and the presence of low and very low channels detecting aerosol plumes at lower altitudes. The LiO3T
exhibits a low relative uncertainty below 20 km, it varies around 69.1 ± 42.7 %. The strong increase above 20 km
is essentially explained by the very low SNR for this instrument at these altitude ranges.

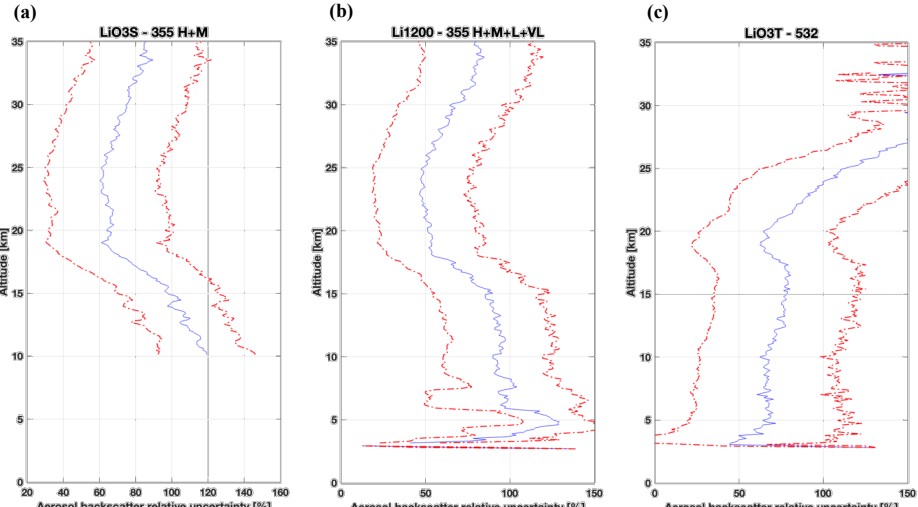


**Figure 13: Mean (blue line) and standard deviation (dotted red line) of the time-series of relative uncertainty from the inversion technique for the (a) lidar O3S (H+M channel), (b) lidar 1200 (H+M+L+VL channel) and (c) lidar O3T (polarized channels).**


**6.  Data availability**
Raw L₀ files, cleaned L₁ files and processed L₂ files with optical products are generated locally. L₀ files are
made of 1minute integrated raw files in licel format. L₁ products contain 1-minute integrated time-series and
overnight averaged cleaned signals in mat file format and netcdf format. L₂ products in mat file format contain
overnight averaged processed signals, as well as range-corrected signals for Raman channels. L₂ products are also



computed in netcdf format following NDACC guidelines in anticipation for a future NDACC label request. **Table**
**C1** in **Appendix C** summarizes the optical products and other variables available in these $L_2$ netcdf files.
Each of these files is available on request in our local datacenter by FTP (ftp://tramontane.univ-reunion.fr/). $L_1$
and $L_2$ files are currently available at https://doi.org/10.26171/rwcm-q370 (Gantois et al., 2024). Mat files and
netcdf files with $L_2$ data will soon be available on AERIS database, but only $L_2$ netcdf files will be openly
accessible.
**7.   Summary**
This study supports the first ever long-term time-series of multiwavelength aerosol optical properties generated
from three lidars operating at the Observatory of Atmospheric Physics of La Réunion (OPAR) since 2013. A full
description of the technical specifications for the three instruments is provided, as well as details about the
preprocessing and processing methods used to produce the different dataset levels. The three time-series consist
in vertical profiles of aerosol elastic backscatter and extinction coefficients at 355 and 532 nm, and linear
depolarization ratio at 532 nm above Maïdo observatory (2160 m asl, west side of Reunion Island, Southern
Hemisphere) from 2013 until now.
The preprocessing step required manual cleaning of more than 1700 files, and the highest frequency of cloud
occurrence resulted in a lower number of validated profiles during the wet season. Data processing methods and
the Klett inversion technique chosen for this work are detailed and referenced. One issue concerns the random
misalignments and technical modifications for the three instruments leading to highly variable parameters such as
the geometrical form factor. As an alternative to the Klett method, the Raman inversion technique has been
attempted but failed for stratospheric and high tropospheric levels due to a poor SNR.
Intercomparison between the three instruments show a good correlation in terms of sAOD values. The
uncertainty analyses reveal a strong influence of the LR value in the low-altitude ranges and a strong influence of
the SNR in the high-altitude ranges. Uncertainty values relative to the total backscatter coefficient are low for the
three instruments. Uncertainty values relative to the aerosol backscatter coefficient are high for the three
instruments because of the very low aerosol backscatter coefficient values generally observed above Maïdo
observatory. Among the three instruments, the LiO3S stands out as the most stable (less misalignments, less
technical modifications) and should be considered the reference instrument at 355 nm. However, data from the
Li1200 can be used to fill the gaps of the LiO3S database and for specific case studies.
**Appendices**
**Appendix A**
The equation describing the desaturated lidar signal can be written as:
$$S_{desat}(\lambda, z) = C(\lambda) \cdot \frac{F(z)}{(z - z_0)^2} \cdot \left\{ \sum_i \beta_i(\lambda, z) \right\} \cdot \left\{ exp\left[ -\frac{2}{\cos(\theta)} \cdot \sum_i \tau_i(\lambda, z_0, z) \right] \right\} + S_{bck}(\lambda) \quad (A1)$$
With $C$ the instrumental constant, $F$ the overlap function, $\beta_i$ the backscatter coefficient of the component $i$, $\tau_i$ the
integrated extinction coefficient of the component $i$ between altitude $z_0$ and $z$, and $S_{bck}$ the background signal.
The range-corrected, sky background corrected and desaturated signal can then be considered:

$$S_2(\lambda, z) = [S_{desat}(\lambda, z) - S_{bck}(\lambda, z)].(z - z_0)^2 \quad (A2)$$


Derivation of the logarithm of $S_2$ leads to:

$$\frac{\delta[ln(S_2)]}{\delta z} = \frac{1}{\beta(\lambda, z)} \cdot \frac{\delta[\beta(\lambda, z)]}{\delta z} - 2.LR_a(\lambda, z).\beta(\lambda, z) - 2.\alpha_m(\lambda, z).\left(1 - \frac{LR_a(\lambda, z)}{LR_m}\right) \quad (A3)$$


With $a$ (resp. $m$) the particular (resp. molecular) contribution, $\alpha(\lambda, z)$ (resp. $\beta(\lambda, z)$) the summed molecular and
particular extinction (resp. backscatter), and $LR$ the Lidar Ratio:
$$LR_a(\lambda, z) = \frac{\alpha_a(\lambda, z)}{\beta_a(\lambda, z)} \quad (A4)$$

$$LR_m(\lambda, z) = \frac{\alpha_m(\lambda, z)}{\beta_m(\lambda, z)} = \frac{8\pi}{3} * K_f \quad (A5)$$

With $K_f$ corresponding to the King factor's value.
The two-component solution of this Bernoulli equation is:

$\beta(\lambda, z) = \beta_a(\lambda, z) + \beta_m(\lambda, z)$
$$= \frac{S_2(\lambda, z).\exp\{2.\int_{z'=z}^{z_{ref}}\left(\frac{LR_a(\lambda, z')}{LR_m(\lambda, z')} - 1\right).\alpha_m(\lambda, z')dz'\}}{\frac{S_2(\lambda, z_{ref})}{\beta(\lambda, z_{ref})} + 2.\int_{z'=z}^{z_{ref}}LR_a(\lambda, z').S_2(\lambda, z').\exp\{2.\int_{x'=z}^{z_{ref}}\left(\frac{LR_a(\lambda, x')}{LR_m(\lambda, x')} - 1\right).\alpha_m(\lambda, x')dx'\}.dz'} \quad (A6)$$


**Appendix B**
The uncertainty budget was determined from the Klett elastic one components inversion technique. Mathematical
details can be found in (Rocadenbosch et al., 2010) for the total backscatter inversion uncertainty budget and
(Sicard et al., 2009) for the two components inversion uncertainty budget.
The Klett inversion was applied to the filtered signal following (see section 4.3.4.):
$$S_{filt}(z) = \frac{S_2(z)}{F(z)} * \frac{coef}{\sum coef} \quad (3)$$

Considering $C = \frac{coef}{\sum coef}$ and $S_{geo}(z) = \frac{S_2(z)}{F(z)}$, the uncertainty of the filtered signal followed the equation:
$$u_{filt}(z) = \sqrt{\left[\frac{\partial S_{filt}(z)}{\partial S_{geo}(z)}.u_{S_{geo}}(z)\right]^2 + \left[\frac{\partial S_{filt}(z)}{\partial C}.u_C(z)\right]^2} = \sqrt{\left[C(z).u_{S_{geo}}(z)\right]^2 + \left[S_{geo}(z).u_C(z)\right]^2} \quad (B1)$$




**Table B1** is a summary of the Total-Backscatter analytical error bars to compute in Klett's backward inversion
method.

| Uncertainty source | Equation |
|---|---|
| **Uncertainty due to the Rayleigh calibration value ($u_{altref}$)** | $u_{altref} = \left\| \left(\dfrac{\beta_j}{\beta_N}\right)^2 \dfrac{U_N}{U_j} \right\| \sigma_{\beta_N}$ |
| **Uncertainty due to the lidar ratio value ($u_{LR}$)** | $u_{LR} = \left\| \pm p\,\dfrac{2\beta_j^2}{U_j} G_j + p^2\,\dfrac{4\beta_j^3}{U_j^2} G_j^{\,2} \right\|$ <br> **Where:** $G_j = \sum_{i=j}^{N} w_i S_i U_i$ |
| **Uncertainty due to the SNR vertical distribution ($u_{SNR}$).** | $u_{SNR} = \sqrt{ \left(\dfrac{\beta_j}{U_j}\right)^2 \sigma_{U_j}^2 + \left(\dfrac{2\beta_j}{U_j}\right)^2 \sigma_{GU_j}^2 }$ <br> **Where:** $\sigma_{GU_j}^2 = \sum_{k=j}^{N}(w_k S_k)^2 \sigma_{U_k}^{\,2}$ |
| **Uncertainty due to the SNR value at the calibration altitude ($u_{SNR,altref}$).** | $u_{SNR,altref} \approx \left\| \dfrac{\beta_j^2}{\beta_N U_j} \right\| \sigma_{U_N}$ |

**Table B1: Total-Backscatter analytical error bars from Klett's backward inversion method (from Rocadenbosch et al.,**
**2010)**

With $\beta_j$ the total backscatter at the altitude cell j, $U_j$ the range-corrected signal at the altitude cell j, N the calibration
altitude cell, $\sigma_{U_j}$ the uncertainty if the range-corrected signal U, $\sigma_{\beta_j}$ the uncertainty of the total backscatter, $S_j$ the
total lidar ratio.
The uncertainty of the total backscatter error bars $u_{\beta T}$ can then be written as:
$$u_{\beta T} = \sqrt{u_{altref}^2 + u_{LR}^2 + u_{SNR}^2 + u_{SNR,altref}^2} \quad (B2)$$




**Appendix C**

| Variable | Dimension | Unit |
|---|---|---|
| **CHANNELS_ID** | channel | - |
| **LATITUDE_INSTRUMENT** | time | deg |
| **LONGITUDE_INSTRUMENT** | time | deg |
| **STATION_HEIGHT** | time | m_asl |
| **DATETIME** | time | MJD2K |
| **DATETIME_START** | time | MJD2K |
| **DATETIME_STOP** | time | MJD2K |
| **INTEGRATION_TIME** | time | h |
| **WAVELENGTH_EMISSION** | channel | nm |
| **WAVELENGTH_DETECTION** | channel | nm |
| **ANGLE_VIEW_ZENITH** | time, channel | deg |
| **ACCUMULATED_LASER_SHOTS** | time, channel | 1 |
| **ALTITUDE** | points | m_asl |
| **AEROSOL_RETRIEVAL_METHOD** | time | - |
| **AEROSOL_BACKSCATTER_RATIO_BACKSCATTER** | time, channel, points | 1 |
| **AEROSOL_BACKSCATTER_RATIO_BACKSCATTER_UNCERTAINTY_COMBINED_STANDARD** | time, channel, points | 1 |
| **AEROSOL_BACKSCATTER_RATIO_BACKSCATTER_RESOLUTION_ALTITUDE_IMPULSE_RESPONSE_FWHM** | time, channel, points | m |
| **RANGE_INDEPENDENT_NORMALIZATION** | time | m_asl |
| **RANGE-CORRECTED_SIGNAL** | time, channel, points | Photons.s-1 |
| **AEROSOL_BACKSCATTER_COEFFICIENT_DERIVED** | time, channel, points | m-1.sr-1 |
| **AEROSOL_BACKSCATTER_COEFFICIENT_DERIVED_UNCERTAINTY_COMBINED_STANDARD** | time, channel, points | m-1.sr-1 |
| **AEROSOL_BACKSCATTER_COEFFICIENT_DERIVED_RESOLUTION_ALTITUDE_IMPULSE_RESPONSE_FWHM** | time, channel, points | m |
| **PRESSURE_INDEPENDENT** | points | hPa |
| **TEMPERATURE_INDEPENDENT** | points | K |
| **AEROSOL_EXTINCTION_COEFFICIENT_DERIVED** | time, channel, points | m-1 |
| **AEROSOL_ EXTINCTION _COEFFICIENT_DERIVED_UNCERTAINTY_COMBINED_STANDARD** | time, channel, points | m-1 |
| **AEROSOL_ EXTINCTION _COEFFICIENT_DERIVED_RESOLUTION_ALTITUDE_IMPULSE_RESPONSE_FWHM** | time, channel, points | m |
| **AEROSOL_LIDAR_RATIO_INDEPENDENT** | time, channel, points | sr |
| **VOLUME_LINEAR_DEPOLARIZATION_RATIO** | time, channel, points | 1 |
| **VOLUME_LINEAR_DEPOLARIZATION_RATIO _UNCERTAINTY_COMBINED_STANDARD** | time, channel, points | 1 |
| **VOLUME_LINEAR_DEPOLARIZATION_RATIO _RESOLUTION_ALTITUDE_IMPULSE_RESPONSE_FWHM** | time, channel, points | m |
| **AEROSOL_LINEAR_DEPOLARIZATION_RATIO_DERIVED** | time, channel, points | 1 |
| **AEROSOL_LINEAR_DEPOLARIZATION_RATIO_DERIVED _UNCERTAINTY_COMBINED_STANDARD** | time, channel, points | 1 |
| **AEROSOL_LINEAR_DEPOLARIZATION_RATIO_DERIVED _RESOLUTION_ALTITUDE_IMPULSE_RESPONSE_FWHM** | time, channel, points | m |

**Table C1: Variables available in the L₂ netcdf files**

Earth System
Science
Data

## Appendix D

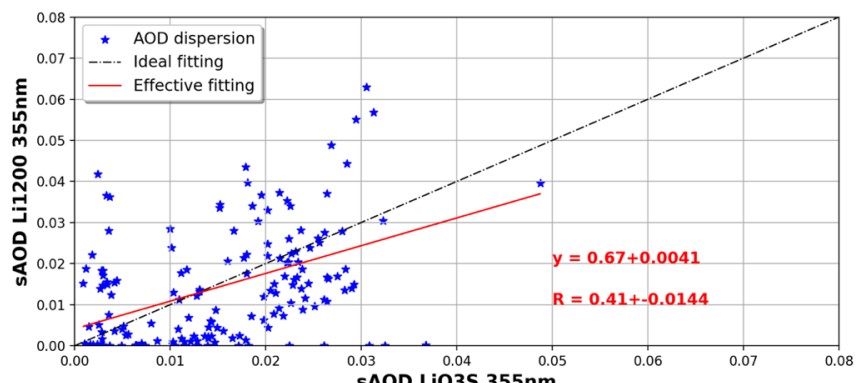

**Figure D1: Dispersion of the AOD (10 to 17 km layer) at 355 nm, between the Li1200 and LiO3S. The black line represents the theoretical linear regression and the red line the actual linear regression.**

**Author contributions.** DG conducted this study with the help of MS, GP, and VD. DG performed the processing of the lidar measurements, the uncertainty analysis, prepared the figures and the manuscript. GP and MS both contributed to the improvement of the text, figures, and uncertainty analysis of this manuscript. GP designed two original softwares used for data processing, which were improved by DG. NM designed the lidar optical schemes. TP and SGB were responsible for the LiO3S instrument and dataset, VD and NM were responsible for the LiO3T instrument and dataset, and VD and GP were responsible for the Li1200 instrument and dataset. PH and EG performed the lidar measurements and the instrumental maintenance and reviewed the technical aspects of this paper. All co-authors contributed to reviewing drafts of this manuscript.

**Competing interests.** The authors declare that they have no conflict of interest.

**Acknowledgments.** The authors gratefully acknowledge Louis Mottet and Yann Hello, who are deeply involved in the routine lidar observations at the Maïdo facility.

**Financial support.** The authors acknowledge the support of the European Commission through the REALISTIC project (GA 101086690). OPAR is presently funded by CNRS (INSU), Météo France, and Université de La Réunion, and managed by OSU-R (Observatoire des Sciences de l'Univers de La Réunion, UAR 3365). OPAR is supported by the French research infrastructure ACTRIS-FR (Aerosols, Clouds, and Trace gases Research InfraStructure – France) and by the French Center for Spatial Studies (CNES). The projects OBS4CLIM (Equipex project funded by ANR: ANR-21-ESRE-0013), EECLAT and AOS (CNES) are acknowledged.

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
