# Peer review of "Multiwavelength, aerosol lidars at Maïdo supersite, Reunion Island, France: instruments description, data processing chain and quality assessment"

_Earth System Science Data, 2024_

## Author Comment (AC2)

|  | Relative Mean Bias Error | Linear Regression Slope | Correlation coefficient |
|---|---|---|---|
| **sAOD** | - 7.59 % | 0.92 | 0.92 ± 0.0047 |
| **Ångström exponent** | - 6.55 % | 0.88 | 0.90 ± 0.1283 |

**Table 2: Intercomparison between the three instruments in terms of sAOD and Ångström exponent.**

| | Li1200 | | LiO3S | LiO3T | |
|---|---|---|---|---|---|
| **Time-series** | In 2013-2017 | 2017-ongoing | 2013-current | In 2013-2017 | 2017-ongoing |
| **Elastic reception channels (nm)** | 355H, 355M | + 355L, 355VL | 355H, 355M | Elastic // 532
Elastic ⊥ 532 | + 532H |
| **Geometrical form factor correction method** | - | From vertical measurements | - | - | - |
| **Polarization calibration method** | - | | - | Rayleigh method | 3 signals method |
| **Inversion method** | Klett | | | | |
| **Vertical resolution range (km)** | 0.030 – 0.522 | 0.015 – 0.522 | 0.030 – 0.261 | 0.052 – 0.522 | |
| **Detection lower bound (km)** | 10 | 3 | 10 | 3 | 10 |
| **Detection upper bound (km)** | 45 | 45 | 40 | 25 | 35 |
| **Uncertainty averaged lowest values (%)** | $50.3 \pm 29.0$ | | $64.4 \pm 31.6$ | $69.1 \pm 42.7$ | |

Table 3: Summary of the processing method and area of validity for the Level 2 products.

---

## Author Response (AR2)

**RESPONSE FILE**

**RC1:**

Very well-written paper describing in necessary detail set of lidar data from which authors and users can extract aerosol information (aerosol optical depth for example). Data easy to find and use. Strong recommendation for publication.

I list a very few issues, below. While well done, this work seems a bit old? Necessary, perhaps, if one wants timescales back to 2013 and decent SNR from a very clean atmosphere. Today, one might start from high spectral resolution lidar (HSRL) data to get AOD directly (authors will undoubtedly know formative work by Eloranta and colleagues)?

*China operates an HSRL from satellite? Several groups in USA fly HSRL on modern aircraft? Perhaps these authors do the best they can with existing lidars? Do any of the HSRL offer validation opportunities?*

Thank you for your comment. Unfortunately, there is no HSRL in Reunion Island. Such a database has never been used until now and it is critical to establish the first long-term time-series and climatology of aerosol optical products in this sparsely documented and highly variable area.

It is true that China operates an HSRL from space: the Aerosol and Carbon Detection Lidar (ACDL) launched in April 2022. The HSR channel is at 532 nm. ESA also launched the satellite EarthCARE in May 2024 which has a HSR channel at 355 nm. In both cases, these spaceborne instruments were not intercompared to our database (which ends in 2023) because of either too low, or no temporal overlap. However, Reunion Island is involved in the upcoming EarthCARE cal/val campaign.

It is true that several US groups, including the NASA Atmospheric Radiation Measurement (ARM) group, and at least one German group, the Deutsches Zentrum für Luft- und Raumfahrt (DLR), operate HSRL. But no field campaign involving such instruments has ever occurred in this part of the world.

*Line 33 (abstract): LR (Lidar ratio) not defined here (not defined until line 420).*

L32 & L33: This correction has been made.

*Line 55: "scarcely influenced by anthropic aerosols"; I think authors mean 'rarely influenced by anthropogenic aerosols'. Language mismatch, not a big issue.*

The impact of anthropogenic aerosols over Maïdo observatory is negligible.

L62: I changed "scarcely" to "barely".

In following lines authors list several factors (biomass burning, volcanism, etc.) that do impact aerosol field. In final sentence of this paragraph (line 63), authors use the term 'quasi-pristine'? Perhaps some attention needed to intent and content of this paragraph. (Fig 9 demonstrates impact of both biomass-induced smoke and volcanic emissions?)

The term quasi-pristine means that, in the absence of any volcanic eruption or biomass-burning plume, the atmosphere composition is particularly clean, because it is free of any anthropogenic pollution.

Two questions:

1. How do authors deal with eye safety at these wavelengths. Never in horizontal operation? No or few external viewers? No or few overflying aircraft? Careful selection of broadcast times? No worries? Perhaps authors should supply a sentence or two to clarify?

Maïdo lidars are large and cannot be moved to make horizontal measurements: the beams of the different lidars are always vertical. To avoid any problems with flying objects, a no-fly zone around the Observatory is requested before each lidar measurement and during operating hours (exclusively nighttime). The research building hosting these instruments has a restricted access. It is located far from any residential areas. The instruments themselves can only be accessed by trained authorized personnel equipped with personal protective equipment (including eye protection glasses for the laser wavelengths) and Optical Enclosures.

L 263: These sentences were added for clarification.

2. Related: have authors considered overflights of aircraft carrying similar lidars plus other aerosol instrumentation?

This would be a very nice way to compare and validate our ground-based measurements. We are thinking about it for future campaigns.

**RC2:**

Gantois et al presented a detailed description of 10-year aerosol optical measurement data at Reunion Island, France, including three active lidars, data protocols, and quality control. The instruments, data processing and quality control are well presented. I understand that these are the key objectives of this manuscript, however, it would be great to include more discussions on the usefulness of the data to meet the interests of broader readers (see major comments below).

Major comments:

*I recommend the authors include more content to present the usefulness and implications of the dataset, some examples that can be explored are:*

- Compare the measurements (aerosol, ozone) during volcanic eruptions vs. non-eruptions. E.g. peak value difference, amplitude change, etc.
- Statistics on 10-year trends and variability of the measurements
- Summary of literature research that used this dataset and the key research questions addressed
- What other research questions can be addressed in the future with this dataset?
- Implications of the measurements of volcanic eruptions on local air quality (e.g. estimate on how much aerosol/ozone was formed or transported to the local during eruptions)

Thank you for this comment. The goal of this work is, in fact, to prepare a quality-assured database using the Maïdo lidar measurements to be later used in geophysical studies where all fields suggested by the reviewer will be exploited. The interest of such a database in Reunion Island is mentioned in the third paragraph of the introduction where the aerosol influence in Reunion Island is described. To inform the reviewer, our team is currently preparing two studies analyzing the variability and trend of the tropospheric and stratospheric time-series (2013-2024) of aerosol optical products. A third study also uses the stratospheric time-series (2022-2023) at 355 nm to analyze the radiative impact of the aerosol and water vapor injected in the stratosphere by the Hunga Tonga eruption.

Our data will also feed NDACC and ACTRIS aerosol remote sensing databases.

In our opinion, and given the scope of your journal, no additional content concerning the application of our data is needed in the revised manuscript. However, should it be of major importance for the reviewer, please tell us and we will add a paragraph highlighting some of the possible studies exploiting the database described in this paper.

Specific comments:

Abstract: please introduce abbreviations when first used, e.g. AOD, SNR, LR

**L27; L32, L33: The corrections have been made.**

L19: add latitude and longitude of the site

L19 & L20: Latitude and longitude have been added in the abstract.

*Table 1: (1) 'Time series'; (2) what does 'added in 2017' mean? If it implies another sensor added to the site from 2017, you can revise it to '2017-ongoing'*

'Time-series': correction made in Table 1.

'2017-ongoing': correction made in Table 1.

Figures 1,2,3: remove red underlines

L146, L194, L244: corrections made.

**Section 2:**

(1) when first mentioning the manufacturers, such as LightMachinery, Hamamatsu, please include a link to the product information page

**Concerning LightMachinery products, we added the following links:**

L95: The emission consists in two Nd: YAG lasers Pro-290, Quanta-Ray Pro Series, from Spectra-Physics, emitting electromagnetic pulses at 1064 nm and 30 Hz (https://www.laserlabsource.com/files/pdfs/solidstatelasersource\_com/product-305/Nd Yag Laser Nanosecond Laser 1064nm 1250mJ Spectra Physics-1462086952.pdf). L158: An excimer laser IPEX-840, PulseMaster PM-800 Series excimer laser with XeCl gas from LightMachinery (https://lightmachinery.com/lasers/excimer-lasers/ipex-800/), emits electromagnetic pulses at 308 nm wavelength with a frequency of 40 Hz and pulse energy of 220 mJ.

L160: A Nd: YAG laser Lab-150, Quanta-Ray Lab Series from Spectra-Physics emits electromagnetic pulse at a 1064 nm wavelength with a frequency of 30 Hz (https://www.laserlabsource.com/files/pdfs/solidstatelasersource\_com/product- 305/Nd Yag Laser Nanosecond Laser 1064nm 1250mJ Spectra Physics-1462086952.pdf).

L210: The emission consists in a Nd: YAG lasers Pro-290, Quanta-Ray Pro Series, from Spectra-Physics, emitting electromagnetic pulses at 1064 nm and 30 Hz (https://www.laserlabsource.com/files/pdfs/solidstatelasersource com/product-305/Nd Yag Laser Nanosecond Laser 1064nm 1250mJ Spectra Physics-1462086952.pdf).

Concerning Hamamatsu products, we bought our detectors from the Licel company. PMTs bought before 2017 were from the R7400 Hamamatsu family, and PMTs bought after 2017 from the R9880 Hamamatsu family. Product information **PMTs** found about can be at https://www.hamamatsu.com/content/dam/hamamatsuphotonics/sites/documents/99 SALES LIBRARY/etd/PMT TPMZ0002E.pdf, and about avalanche

photodiodes at https://www.hamamatsu.com/content/dam/hamamatsu- photonics/sites/documents/99 SALES LIBRARY/ssd/si apd kapd0001e.pdf.

We added these links L130, L182 and L226.

(2) it would be good to have a table summarizing calibration processes, instrument precision, detection limits, uncertainty.

As requested by the reviewer, the following Table 3 provides a summary of the processing method and area of validity for the Level 2 products.

|                                           | Li1200 LiO3S  |                            | LiO3T              |                 |                  |
|-------------------------------------------|---------------|----------------------------|--------------------|-----------------|------------------|
| Time-series                               | In 2013-2017  | 2017-ongoing               | 2013-current       | In 2013-2017    | 2017-ongoing     |
| Elastic reception channels (nm)           | 355H, 355M    | + 355L, 355VL              | 355H, 355M         | Elastic // 532  | + 532H           |
|                                           |               |                            |                    | Elastic ⊥ 532   |                  |
| Geometrical form factor correction method | -             | From vertical measurements | -                  | -               | -                |
| Polarization calibration method           | -             |                            | -                  | Rayleigh method | 3 signals method |
| Inversion method                          | Klett         |                            |                    |                 |                  |
| Vertical resolution range (km)            | 0.030 - 0.522 | 0.015 - 0.522              | 0.030 - 0.261      | 0.052 - 0.522   |                  |
| Detection lower bound (km)                | 10            | 3                          | 10                 | 3               | 10               |
| Detection upper bound (km)                | 45            | 45                         | 40                 | 25              | 35               |
| Uncertainty averaged lowest values (%)    | 50.3 ± 29.0   |                            | 64.4 ± 31.6 | 69.1 ± 42.7     |                  |

 Table 3: Summary of the processing method and area of validity for the Level 2 products.

This Table was added to the manuscript L635.

L244: I don't understand this sentence 'This is xxx'

L297: This sentence was inexact and removed.

*L245: what is 'signal enabled'?*

L297: This sentence was changed to 'The strong disturbance in the signal made it easy to fully automatize their detection'.

*L408: why use different methods for the two time periods?*

L467: We used different methods for the two time-periods because a 532H channel was added in 2017 which enabled us to compare total and polarized channels. This method leads to a smaller uncertainty compared to the Rayleigh calibration method.

Section 5.3: significant tests are needed for comparing two instruments

In this section, the comparison is made for the sAOD at 355 nm from the LiO3S and Li1200. The performance of both datasets is assessed in terms of correlation slope and correlation coefficients, both in sAOD and in AE (by including the wavelength of 532 nm of a single instrument, the LiO3T). We added calculation of the relative Mean Bias Error (MBE) for the two variables and summarized these metrics in the following Table 2.

|                   | Relative Mean Bias Error | Linear Regression Slope | Correlation coefficient |
|-------------------|---------------------------------|-------------------------|--------------------------------|
| sAOD              | - 7.59 %                        | 0.92                    | $0.92\pm0.0047$                |
| Ångström exponent | - 6.55 %                        | 0.88                    | $0.90 \pm 0.1283$              |

Table 2: Intercomparison between the three instruments in terms of sAOD and Ångström exponent.

This analysis shows a good agreement between the two datasets. After identifying the LiO3S as the reference instrument at 355 nm, we found a negative MBE concerning sAOD, meaning that the Li1200 tends to underestimate sAOD compared to LiO3S.

L569: Table 2 was added along with the following sentences:

"Table 2 summarizes the metrics used to intercompare the three instruments. The relative Mean Bias Error (MBE) was added the analysis. After identifying the LiO3S as the reference instrument at 355 nm, we found a negative MBE (- 6.55 %) concerning sAOD, meaning that the Li1200 tends to underestimate sAOD compared to LiO3S."

L503: it would be good to plot the time series of LiO3T

We chose not to plot the LiO3T time-series, as there are already many illustrations in this paper. However, it will appear in the following paper dedicated to the geophysical analysis of the aerosol optical products in the troposphere.

**New contributing author**

L5 & L727: Nelson Bègue was added to the list of authors. He valuably contributed during the reviewing phase of the manuscript.

**Other modifications**

L862: The reference of the preprint article "Radiative impact of the Hunga Tonga-Hunga Ha'apai stratospheric volcanic plume: role of aerosols and water vapor in the southern tropical Indian Ocean," was updated.